# Role of Mitochondria-Cytoskeleton Interactions in the Regulation of Mitochondrial Structure and Function in Cancer Stem Cells

**DOI:** 10.3390/cells9071691

**Published:** 2020-07-14

**Authors:** Jungmin Kim, Jae-Ho Cheong

**Affiliations:** 1Brain Korea 21 PLUS Project for Medical Science, Yonsei University College of Medicine, Seoul 03722, Korea; jm_kim@yuhs.ac; 2Department of Surgery, Yonsei University Health System, Yonsei University College of Medicine, 50 Yonsei-ro, Seodaemun-gu, Seoul 03722, Korea; 3Yonsei Biomedical Research Institute, Yonsei University College of Medicine, Seoul 03722, Korea; 4Department of Biochemistry & Molecular Biology, Yonsei University College of Medicine, Seoul 03722, Korea; 5Department of Biomedical Systems Informatics, Yonsei University College of Medicine, Seoul 03722, Korea

**Keywords:** cytoskeleton, tubulin, cytoskeleton-mitochondria, cancer stem cell, cancer mitochondria, cancer metabolism, cancer energy metabolism

## Abstract

Despite the promise of cancer medicine, major challenges currently confronting the treatment of cancer patients include chemoresistance and recurrence. The existence of subpopulations of cancer cells, known as cancer stem cells (CSCs), contributes to the failure of cancer therapies and is associated with poor clinical outcomes. Of note, one of the recently characterized features of CSCs is augmented mitochondrial function. The cytoskeleton network is essential in regulating mitochondrial morphology and rearrangement, which are inextricably linked to its functions, such as oxidative phosphorylation (OXPHOS). The interaction between the cytoskeleton and mitochondria can enable CSCs to adapt to challenging conditions, such as a lack of energy sources, and to maintain their stemness. Cytoskeleton-mediated mitochondrial trafficking and relocating to the high energy requirement region are crucial steps in epithelial-to-mesenchymal transition (EMT). In addition, the cytoskeleton itself interplays with and blocks the voltage-dependent anion channel (VDAC) to directly regulate bioenergetics. In this review, we describe the regulation of cellular bioenergetics in CSCs, focusing on the cytoskeleton-mediated dynamic control of mitochondrial structure and function.

## 1. Introduction

Mitochondria orchestrate diverse fundamental cellular functions, including respiration, calcium homeostasis, reactive oxygen species generation, and programmed cell death [1,2,3,4]. They have been implicated in multiple steps of tumor progression and maintenance of cancer stemness, and heterogeneity in the morphology and spatial distribution of these organelles in cancer is becoming a field of intense investigation [3,5]. The relationship between cancer stem cells (CSCs) and mitochondria has been recently demonstrated to be an important target for cancer treatment [6,7,8,9]. Determining the mitochondrial regulatory mechanisms in order to understand CSC bioenergetics and the role of mitochondria is important in studies of cancer biology. Components of the cytoskeleton play a vital role in the structural and functional organization of the mitochondria, including mitochondrial morphology, dynamics, motility, and intracellular arrangement [5,10]. Additionally, the interaction of mitochondria with cytoskeletal components and their interactions with voltage-dependent anion channels (VDACs) are involved in the coordination of mitochondrial function [11,12,13,14,15]. Moreover, alterations in the cytoskeletal architecture occur to meet the need for cellular energy during proliferation, apoptosis, and adaptation to challenging conditions, such as the lack of an energy source, extrinsic and intrinsic stresses, and mechanical stress [16]. Modulation of mitochondrial dynamics and cytoskeletal features can profoundly impact CSC behavior. Significant efforts in recent years have revealed that mitochondrial function is critical for CSC survival [6,8,9,17]. Although mitochondrial function has now been implicated in multiple CSCs, the underlying mechanisms regulating the mitochondrial architecture and bioenergetics are not well understood. Numerous reports have demonstrated the relationship between the cytoskeleton and CSCs in the context of epithelial-mesenchymal transition (EMT) and metastasis because of the essential role of the cytoskeleton in the maintenance and regulation of cellular architecture [18,19]. Yet, the relationship between the cytoskeleton and CSCs focused on mitochondrial functional augmentation, which can provide CSCs with a survival advantage is unclear. In this review, we summarize previous studies providing evidence for the role of cytoskeletal components in regulating the mitochondrial structure, function, and bioenergetics in a CSC model system. An in-depth understanding of CSC behavior can provide insights into cancer biology that may improve clinical applications aimed at targeting the bioenergetics of CSCs for treating drug-resistant malignant tumors.

## 2. Unique Features of CSCs

CSCs are more resistant to chemotherapeutic agents compared to bulk tumor cells, and therefore survive conventional therapies, resulting in relapse of the disease. From a bioenergetics perspective, CSCs exhibit mitochondria-centric metabolism and have the capability to adapt to harsh conditions like lack of energy sources, and can thus survive and maintain their stemness features.

### 2.1. Identification of CSCs

In recent years, CSCs have gained interest as key tumor-initiating cells that may also play an integral role in tumor recurrence and chemoresistance. Identification and isolation of these CSCs using putative surface markers has been a priority in cancer research. The main markers used for isolation and identification of CSCs include surface cell-adhesion molecules (e.g., CD44, CD133, and EpCAM) and cytoprotective enzymes (such as aldehyde dehydrogenase, ALDH). However, the surface markers that are used to isolate or identify CSCs from diverse malignant tissues are expressed not only by CSCs, but also by many normal tissues [20,21,22,23,24]. Fox et al. revealed that normal human tissues express different CD44 isoforms; for instance, normal epithelial tissues and some tumors express a wide range of variants at high levels. The association of putative markers of CSCs such as CD44 and CD133 with drug resistance has been largely based on studies demonstrating reduced sensitivity of CSC marker-positive cells to chemotherapeutic drugs or increased expression of these markers in treatment-resistant tumors [22,25,26]. Elevated ALDH1A1 is observed in CSCs of multiple cancer types [27,28,29,30,31]. Thus, the use of surface marker expression alone is insufficient to identify CSCs. Evaluation of surface markers must be associated with other functional assays, such as sphere-forming assays in a low attachment surface, and measurement of the expression of specific CSC-related genes to provide persuasive evidence for the presence of CSCs. In recent years, mitochondrial and energy metabolism-related indicators of CSCs have been studied by an increasing number of researchers. According to these studies, CSCs are less glycolytic, produce less reactive oxygen species (ROS), and maintain higher ATP levels than bulk tumors. Moreover, the mitochondria in CSCs have increased mass and membrane potential, which is a reflection of augmented mitochondrial function, and enhanced oxygen consumption rates compared with bulk tumors, which generate their energy mainly via glycolysis [32,33,34,35,36,37,38]. Increased mitochondrial mass indicates stem-like phenotypes and is associated with metastatic potential and insensitivity to DNA damage [39]. Taken together, mitochondrial features might be regarded as a useful tool and a novel strategy to identify CSCs, together with the classic CSC markers.

### 2.2. Cellular Bioenergetics in CSCs

The metabolic characteristics of CSCs have been the subject of intense investigation over the past years. Due to the abnormally controlled proliferation and inadequate blood vessel formation as the tumor progresses, tumor microenvironments are characterized by hypoxic, acidic, and nutrient-poor conditions; therefore, cancer cells must adapt their cellular bioenergetics efficiently to deal with these unfavorable conditions, a process called metabolic reprogramming [40]. Metabolic reprogramming is crucial to sustain proliferation and survival of tumor cells when the oncogenic cue is blocked [6,41]. From a bioenergetics perspective, cancer cells exhibit an aerobic glycolytic metabolism known as the Warburg effect [42], whereas CSCs display distinct metabolic features, known as oxidative phosphorylation (OXPHOS). Metabolic plasticity may allow cancer cell subpopulations to switch between glycolysis and OXPHOS to support their bioenergetic needs and ultimately to survive during and after anticancer treatments [43,44,45,46]. In agreement with the concept of metabolic plasticity of cancer cells, JARID1B^high^ cells [8] revealed a prominent increase in oxygen consumption, resulting in upregulation and downregulation of genes involved in the electron transport chain (ETC) and glycolysis, respectively [1]. In surviving cells after KRAS inactivation, transcriptome analysis revealed that expression of genes involved in mitochondrial function, such as ETC and β-oxidation, is increased, and mitochondria have a larger size and consume more oxygen than bulk pancreatic tumor cells [6]. Furthermore, MYC- and MCL1-positive cells show increased OXPHOS with higher bioenergetic reliance on fatty acid oxidation in triple negative breast cancer (TNBC) [17]. Elevated expression of PGC1-α in melanoma is associated with increased dependency on OXPHOS and decreased glycolytic features [47]. Several studies in diverse cancer types, such as ROS^low^ quiescent leukemia stem cells, CD133^+^ cells of glioblastoma, and pancreatic ductal adenocarcinoma, show that they exhibit a preference for OXPHOS rather than glycolysis [33,36,48,49]. OXPHOS is also upregulated as a result of chemotherapy [50]. Increased OXPHOS may support drug resistance by producing sufficient ATP for the removal of drugs via multidrug transporters in a colon cancer model [51]. Oxaliplatin and fluorouracil (5-FU) treatment of colon cancer cells increase OXPHOS through sirtuin 1 (SIRT1) and PGC1-α induction [52]. Furthermore, downregulation of OXPHOS via SIRT1/PGC1-α knockdown resensitizes cancer cells, suggesting that OXPHOS inhibition may represent an attractive addition to adjuvant chemotherapy. Despite mitochondrial function being high in these studies, ROS levels are significantly lower in CSCs [53], which also explains their more powerful antioxidant defense system compared with bulk tumors. A strong quenching process keeps ROS levels low, and helps in the maintenance of the stemness of CSCs, thereby contributing to therapy resistance [54]. According to these observations, understanding the metabolic characteristics of CSCs would be very helpful in designing therapeutic approaches that selectively target CSCs. Furthermore, an effort to investigate the regulatory mechanisms that control cellular bioenergetics of CSCs should provide new therapeutic insights.

## 3. Mitochondrial Structure and Function

Cellular bioenergetics is the foundation of survival for all organisms, including cancer cells. Mitochondria allow cells to sense cellular stress and to adapt to challenging extrinsic or intrinsic stimuli, conferring a high degree of plasticity to tumor cells for growth and survival.

### 3.1. Mitochondrial Respiratory Chain

Electrons derived from different metabolic processes are involved in a series of transmembrane protein complexes in the mitochondrial inner membrane (known as the ETC) to fuel the OXPHOS process. Nicotinamide adenine dinucleotide (NADH), flavin adenine dinucleotide (FADH_2_), and succinate act as electron donors. The ETC is responsible for the transfer of electrons from reduced NADH and FADH_2_, produced in the tricarboxylic acid (TCA) cycle, to oxygen across complexes I, II, III, and IV. This electron flow provides energy for proton translocation, known as the proton motive force [55,56], thereby generating ATP. Oxygen acts as the terminal electron acceptor (Figure 1). The malate-aspartate shuttle (MAS) is indispensable for the transfer of cytosolic NADH into mitochondria to maintain a high rate of glycolysis and to support tumor cell growth. Moreover, the malate-aspartate shuttle has been reported to oxidize 20–80% of cytosolic NADH in diverse tumor cells [57,58]. The glycerol-3-phosphate shuttle (G3P-DHAP), composed of two distinct enzyme components: (1) NAD^+^/NADH-linked glycerol 3-phosphate dehydrogenase, which is cytosolic, and (2) flavin adenine dinucleotide (FAD/FADH_2_)-linked glycerol 3-phosphate dehydrogenase, which is localized to the mitochondrial inner membrane. Unlike the MAS, which transports electrons from cytosolic to mitochondrial NADH, the G3P-DHAP shuttle transfers the electrons directly to ubiquinones that enter the electron transport complex III. Using an intact human platelet study, cell-permeable succinate prodrug (NV118) increased mitochondrial oxygen consumption with energy production in cells intoxicated with complex I inhibitor, metformin [59]. This concept was interpreted as a “bypass” theory. NV118 entered platelets and released succinate; succinate directly donated electrons to complex II and thus reactivated the electron flow to the distal part of the respiratory chain independent of complex I. This result highlights the potential therapeutic application of succinate in conditions accompanied by complex I inhibition. Fatty acids have a higher energy density than glucose, and CSCs attempt to maximize fatty acid utilization under glucose starvation conditions. Several studies that target the mitochondrial complex I might be insensitive in certain cancer cells, which indicates a high degree of redundancy in the survival mechanism of CSCs.

### 3.2. Mitochondrial Dynamics

Mitochondria continually fuse and divide, and their quality, distribution, size, and motility are finely tuned [60,61]. Mitochondrial fission and fusion are known to play roles in maintaining the integrity of mitochondria, electrical, and biochemical connectivity, and protection of mitochondrial DNA (mtDNA) [62,63,64]. Mitochondrial dynamics are regulated by highly conserved large guanosine triphosphatases (GTPases). Fusion of the outer membrane of mitochondria (OMM) is mediated by mitofusin 1 and mitofusin 2 (Mfn1 and Mfn2) [65,66,67], whereas the inner membrane of mitochondria (IMM) is regulated by optic atrophy 1 (OPA1) [68]. Mitochondrial fission is regulated by dynamin-related protein 1 (DRP1) [69] and mitochondrial fission factor (Mff) [70]. An imbalance between mitochondrial fission and fusion leads to dysfunctional mitochondria and is a key feature of diverse disorders, including cancer [71].

Mitochondrial fusion and fission form an essential axis of mitochondrial quality control. However, quality control may not be the only task carried out by mitochondrial dynamics. Depending on the physiological conditions, the mitochondrial fission–fusion balance may be perturbed and mitochondria may appear fragmented, such as during apoptosis and cell division [2], or elongated, such as during nutrient starvation [32,72] (Figure 2). Mitochondrial fission is increased during the G2/M phase of the cell cycle to guarantee an accurate mitochondrial segregation between the two daughter cells during cell division [73], and mitochondrial fusion prevents the removal of mitochondria by starvation-induced autophagy. Mitochondrial fission is crucially involved in maintenance of healthy mitochondria populations, including mitochondrial inheritance by daughter cells during cellular division. Mitochondrial fragmentation occurs in response to various factors such as intracellular calcium levels and ATP availability [74,75]. For example, the calcium-sensitive phosphatase calcineurin promotes fission by dephosphorylating cytosolic Drp1, which causes translocation to mitochondria [76]. Plasticity bestows the adaptive flexibility needed to adjust and adapt to changing cellular stresses and metabolic demands. In respect to the plasticity of mitochondrial dynamics, we have suggested that constant network-like remodeling establishes a mechanism for quality control of the mitochondrial population with important ramifications for the gain fitness of CSCs. Mitochondrial elongation is necessary to maintain structural and functional homogeneity, to preserve mitochondrial genome integrity, and to ensure the proper balance between energy generation and cellular mass [77,78,79,80,81,82]. Mitochondrial fusion causes an increase in the mitochondrial cristae number, which is associated with dimerization of the ATP synthase and thereby upregulates ATP synthesis activity [72]. Furthermore, this mitochondrial inner membrane rearrangement is involved in cristae remodeling and protection from cytochrome c release [83,84], suggesting that mitochondrial fusion is important for the maintenance of healthy mitochondria. Recent reports demonstrated that mitochondrial fusion during nutrient starvation facilitates fatty acid trafficking and oxidation for survival [85,86]. Upon starvation, cells remodel mitochondria into highly connected networks and also upregulate enzymes required for mitochondrial fatty acid trafficking, import and β-oxidation [56]. Taken together, these results show that mitochondrial morphological dynamics and integrity contribute to changes in mitochondrial function, such as utilization of specific energy sources during nutrient starvation.

### 3.3. Mitochondrial Structure-Function in CSCs

Although the underlying mechanisms regulating mitochondrial dynamics in CSCs remain unknown, a number of reports have demonstrated that mitochondrial structure is directly related to mitochondrial function. The importance of mitochondrial morphology and function in CSCs is emphasized by the findings of Viale et al. [6]. The authors have shown that oncogene ablation-resistant, surviving cells in pancreatic cancer exhibit elevated mitochondrial bioenergetic capacity along with highly elongated mitochondria. In a similar manner, neuroblastoma cells with increased resistance to apoptosis upon cisplatin treatment show high expression levels of MFN1 following hyperfused mitochondria and increased mitochondrial respiratory capacity [67]. Studies using stress-resistant pancreatic cancer cells demonstrated that suppression of glycolytic capacity dramatically changed mitochondrial morphology; in particular, mitochondrial fusion was more abundant in cells cultured in low-glucose medium [87]. Recent studies have found that mitochondrial fusion and enhanced oxygen consumption capacity are crucial for regulating and maintaining stemness. For example, MIM fusion mediated by OPA1 is critical for regulating tight cristae junctions and the proximity of ETC complexes to each other in memory T cells compared to effector T cells [88].

Although the underlying mechanisms regulating mitochondrial dynamics in CSCs remain unknown, numerous studies have revealed that hyper-activated oncogenic pathways act as potent signals to remodel the mitochondrial structure and metabolism. As specific mitochondrial morphologies are linked to different energetic states of the cells, oncogenic-signaling-mediated mitochondrial structure may induce changes in mitochondrial function to support altered metabolism. For example, KRAS-ablated surviving cells in pancreatic ductal adenocarcinoma (PDAC) displayed fused mitochondria and relied on OXPHOS [6], whereas B-RAF^V600E^-driven melanoma cells exhibited a fragmented mitochondrial network with upregulated DRP1 expression and increased glycolytic metabolism [89]. Although tumorigenesis can be initiated through oncogenic MAPK or MYC signaling, these pathways result in different mitochondrial phenotypes. Oncogenic MAPK promotes mitochondrial fission, whereas MYC induces mitochondrial fusion. Oncogenic MYC induces the expression of genes involved in cellular metabolism, including diverse genes that regulate mitochondrial mass, and biogenesis results in an increase of metabolic capacity of cancer cells [90]. Knockout of MYC in mouse embryonic fibroblasts (MEFs) exhibited fragmented mitochondria, while re-expression of MYC induces fusion by elevation of OPA1 and MFN2 [91]. In contrast, elevated MAPK signaling amplifies downstream kinase cascades that culminate in increased ERK pathway, immediate activation of DRP1, inactivation of MFN1, and mitochondrial fission [89,92]. In T-cell acute lymphoblastic leukemia cells with acquired drug resistance, ERK activation-induced DRP1 phosphorylation was found to result in pro-glycolytic phenotype switching and mitochondrial fragmentation [93]. Downregulation of mitochondrial fusion proteins reduces OXPHOS and ATP production in diverse cancer models [79,94,95]. In a more complex study, Vaseva et al. found that mutant KRAS regulates MYC via ERK1/2-dependent and independent mechanisms in pancreatic cancer. ERK1/2 inhibition activates a compensatory EGFR-SRC-ERK5 cascade that stabilizes MYC, and combined ERK1/2 and ERK5 inhibition promotes synergistic loss of MYC and suppresses PDAC growth [96]. Studies are needed to test these mechanisms in various model systems to fully understand the role of the oncogenic signaling network in the regulation of mitochondrial structure and function in CSCs. These opposing effects mediated by oncogene activation may be modulated by different capacities of utilization of nutrients from the nutrient-poor tumor microenvironment. As we mentioned above, cancer cells generally rely on glycolysis as the major energy source, whereas CSCs exhibit mitochondrial-centric metabolism, i.e., OXPHOS [97]. Recent reports have suggested that ERK1/2 signaling is necessary to induce glycolysis. The authors’ proposed mechanism for ERK regulation of glycolysis involves the phosphorylation of phosphoglycerate kinase 1 (PGK-1) by ERK1/2 on Ser 203. This phosphorylation induces mitochondrial PGK-1 to act as a protein kinase that phosphorylates and activates PDH kinase 1 (PDHK1) on Thr338. PDHK1, in turn, phosphorylates PDH, inhibiting enzyme activity, and induces glycolysis [98]. In contrast, ATP levels in MYC-positive cells are supported by OXPHOS to a greater extent than in MYC-negative cells [99]. MYC also promotes utilization of TCA cycle intermediates, including fatty acids from acetyl-CoA and NADPH for malic enzymes [100]. In addition, MYC-overexpressing TNBC exhibited elevated fatty acid oxidation and its inhibition suppressed tumorigenesis in vivo [101].

Greer et al. showed that the small-molecule drug ONC201 [102], originally shown to induce the transcription of tumor necrosis factor (TNF)-related apoptosis-inducing ligand (TRAIL) and to destroy breast cancer cells by activating TRAIL death receptors, induces phosphorylation of AMP-dependent kinase and ATP loss. Cytotoxicity and ATP depletion were significantly enhanced under glucose-limiting conditions, suggesting that ONC201 targets mitochondrial respiration. ONC201 inhibits mitochondrial respiration and induces mitochondrial structural damage (e.g., fragmented mitochondria and matrix-lysed mitochondria). Furthermore, Graves et al. found that caseinolytic protease P (ClpP) is a specific binding protein [103]. ClpP is encoded by a nuclear gene, translated in the cytoplasm and then imported into the mitochondria matrix, which plays a central role in mitochondrial protein degradation within the mitochondria in a process known as mitochondrial proteolysis. In addition, ClpP interacts with mitochondrial respiratory chain proteins and metabolic enzymes. Genetic knockdown of ClpP impairs OXPHOS and complex II [104]. Over the past years we have improved our understanding of how mitochondrial dynamics are regulated, as well as cell-specific contexts in which different mitochondrial morphologies are favored. These studies suggest that CSCs exhibit highly fused mitochondrial networks as means of maintaining OXPHOS, ATP production, and also their stemness.

## 4. Cytoskeleton

The cytoskeleton is a system of filaments and fibers that are essential for survival and diverse cellular processes in both normal and cancer cells [105,106]. Alterations in cellular structure and intracellular-organelle rearrangement and repositioning lead to changes in cellular metabolism and the acquisition of inappropriate migratory and invasive features, and also accompany the progression of cancer [107]. Actin and the microtubule cytoskeleton are key modulators that underpin these cellular processes. Ultrastructural analysis in diverse model systems have shown that the cytoskeleton is associated with intracellular membranes and organelles [108,109]. It has been postulated that these linkages represent a regulatory process of the distribution, positioning, and trafficking of mitochondria towards energy demanding areas inside cells. An important question arises: how do cells accomplish these processes?

### 4.1. Role of Actin Microfilaments in the Regulation of Mitochondria

Microfilaments, also called actin filaments, are composed of three isoforms, α-, β-, and γ-actin [110]. Actin exists in two forms: G-actin (globular actin) and F-actin (fibrillar actin). Actin reorganization occurs during cell membrane protrusion and retraction in stress fibers, contractile ring formation during cell division, and mitochondrial fission [111,112,113,114,115]. Several reports have suggested that mitochondria may move along actin filaments in dendrites and axons [116,117,118]. Actin filaments are needed to support microtubules in short-distance trafficking of mitochondria in microtubule-poor regions of the cell [63]. Using live-cell imaging, Moore et al. demonstrated that the association of actin filaments with mitochondria is transient, with rapid disassemble-reassemble cycling throughout the mitochondria within 14 min. In the actin disassemble phase, fragmented mitochondria undergo rapid fusion, which leads to quick local recovery of mitochondrial integrity and maintenance of mitochondrial homeostasis [119]. Moreover, these authors found that F-actin recruitment to mitochondria is dependent on Arp2/3, which has been identified as a component of the mitochondrial outer membrane (MOM). Mitochondrial fission is initiated by marking the division site by the endoplasmic reticulum (ER) tubules, followed by membrane fission via Drp1. The ER-bound inverted formin 2 (INF2), which stimulates polymerization of F-actin at the ER-mitochondria binding sites, leads to mitochondrial fission [115,120,121]. INF2-dependent actin polymerization induces constriction of both mitochondrial membranes during the fission process: IMM constriction is stimulated by increased Ca^2+^ transfer from ER to mitochondria, and OMM constriction is enhanced by increased recruitment of Drp1. MYO6 is an emerging regulator in cytoskeletal dynamics and cancer progression. MYO6 has been used as a marker of an aggressive phenotype in ovarian cancer. Intriguingly, MYO6 triggers F-actin cage assembly around dysfunctional mitochondria, thereby preventing the spread of damaged mitochondria [122]. These reports suggest that the actin cytoskeleton is essential for spatio-regional trafficking, dynamics, and also quality control of mitochondria.

### 4.2. Role of Microtubules in the Regulation of Mitochondria

In eukaryotes, microtubules are one of the major components of the cytoskeleton and participate in many vital processes such as mitosis, structural support, intracellular transport of vesicles, and organelles like mitochondria [105,106,123,124,125]. Microtubules are constantly modified in different patterns to enhance their functions. One type of modification is acetylation on lysine 40 of α-tubulin, which results in acetylated α-tubulin (Ac-AT), which recruits motor proteins to facilitate movement of vesicles and mitochondria along its tracks [126,127,128,129,130,131]. Microtubule-associated motor proteins include kinesin and dynein. Dyneins and kinesins transport their cargo toward the minus and plus ends of microtubules (MT), respectively. In neuronal studies, kinesins and dyneins transport mitochondria along Ac-AT throughout axonal and dendritic cells towards the high energy consumption area (e.g., groups of ion pumps or cell protrusion zones) to generate and distribute ATP [132,133,134]. Dynein has been reported to be associated with mitochondria, possibly via interaction with the outer membrane protein VDAC [135]. It has been noted that, in vitro, loss of acetylated residues reduces the interaction of kinesin with MT, with a subsequent decrease in motility of the motor protein [126,127]. Trafficking of mitochondria along MT tracks is regulated by second messengers generated from signaling events. Several studies have demonstrated that intracellular Ca^2+^, when elevated, halts MT-based mitochondrial movement in many cell types [63,134,136,137]; however, the rapid axonal transport of other cargo persists in the presence of Ca^2+^ [138]. Ca^2+^ influx occurs in areas of high metabolic demand, such as nerve terminals, protrusion zones, and the leading edge of pro-metastatic regions, where energetically valuable mitochondria are clustered [5,139]. Furthermore, sustained elevated cytosolic Ca^2+^ may be a symptom of insufficient local ATP to pump Ca^2+^ out across the plasma membrane, or a defective buffering capacity of mitochondria, owing to decreased activity of the ETC. Intracellular Ca^2+^ level-dependent inhibition of mitochondrial motility occurs in response to spatial and temporal requirements and in order to enhance local Ca^2+^ buffering. In a previous report, we showed that CSCs express sarco/endoplasmic reticulum Ca^2+^-ATPase to avoid apoptosis, which occurs after Ca^2+^ overload under glucose deprived conditions [140]. How Ca^2+^-dependent mitochondrial movement in CSCs is accomplished is presently not well understood. Taken together, the microtubule cytoskeleton is an essential component of mitochondrial trafficking and anchoring through Ca^2+^ regulation.

## 5. Cytoskeleton-Mitochondria Interaction

In addition to regulating mitochondrial dynamics, several pieces of evidence demonstrate that the cytoskeletal network interacts with mitochondria to control mitochondrial function [15,108,109].

### 5.1. Interaction and Regulation of Mitochondrial Function by Actin

Among the most abundant proteins in the cell, actin can self-assemble into polymers (F-actin). As a component of the cytoskeleton, it has the property of interacting with diverse proteins such as the voltage-dependent anion-selective channel (VDAC) and MtCK. VDAC is the most abundant protein of the MOM. It has an important role in the energy management of the cell by regulating the metabolic fluxes and is involved in diverse cellular processes. VDAC acts as a channel or as an anchoring protein and interacts with actin filaments. G-actin, not F-actin, induces the closure of VDAC in vitro [141]. Overexpression of mitochondrial creatine kinase (MtCK) has been reported in several tumors with poor prognosis [142]. Furthermore, increased levels of MtCK in malignant cells may be part of a metabolic adaptation of cancer cells to increase growth rates under oxygen- and glucose-restricted conditions. Upregulated MtCK levels could help to sustain a high energy turnover, but would also be protective against stress conditions such as hypoxia and eventually apoptotic death [143]. Guzun et al. reported that the regulation of energy fluxes is dependent on the interaction of mitochondria with cytoskeletal proteins. The results show that MtCK, which localizes with myosin surrounded by actin filaments in cells, limit VDAC permeability, mostly decreasing ATP and ADP levels. Strongly decreased permeability of MOM for adenine nucleotides significantly enhances the functional coupling between MtCK and ANT (adenine nucleotide translocator), increasing the rate of ADP and ATP recycling in the mitochondrial matrix inner membrane space [144]. Actin, on its own, is suggested to be a regulator of mitochondrial function via interactions with VDAC and MtCK.

### 5.2. Interaction and Regulation of Mitochondrial Function by Microtubules

Microtubules have long been known to interact strongly with mitochondria in many cell types [145]. A growing body of evidence suggests that the microtubule network may interact with mitochondria to control mitochondrial respiration [15,108,109]. Co-localization of β-tubulin with mitochondria and its association with the MOM was demonstrated first by Saetersdal et al. [146]. Additionally, the presence of tubulin in mitochondria has been shown by Carre et al. in different cell types, where tubulin is localized at the MOM. Several studies suggested that ‘mitochondrial’ tubulin can be organized in α/β-dimers, and, using immunoprecipitation, the authors found that this tubulin is associated with the mitochondrial VDAC [15,146,147,148,149,150]. The VDAC controls diverse cellular processes such as cellular metabolism and apoptosis. The VDAC forms channels on the MOM, where the opening/closure of the channel is critical for controlling the permeability of the mitochondrial membrane and the flux of metabolites into the organelle [150]. Mitochondrial tubulin interacts with the VDAC, and nano-molar concentrations of α β-heterodimers can lead to closure of the VDAC [15,148]. In vitro measurements showed the insertion of tubulin’s polyanionic C terminus into the lumen of the VDAC β-barrel as the mechanism modulating the flow of ATP and ADP through the VDAC and regulating metabolite fluxes across the MOM [127,128]. Several studies have shown that the rate of mitochondrial respiration in isolated mitochondria in vitro is strictly regulated by the availability of ADP for ANT in the mitochondrial inner membrane (MIM) [151]. In isolated mitochondria, the MOM is permeable to metabolites with molecular mass lower than 7 kDa, owing to the open state of the VDAC in MOM [133], and the efficiency of mitochondrial function regulation in vitro by extra ADP depends only upon the affinity of ANT for ADP, which is very high. In cancer cells, the closure of the VDAC by tubulin leads to decreased mitochondrial membrane potential, which could alter mitochondrial metabolism [11,152]. Collectively, microtubules regulate mitochondrial function via interaction with mitochondrial outer membrane components. However, many aspects related to the mitochondria–tubulin interplay in CSCs have yet to be elucidated. In particular, revealing the precise nature and possible functional roles of various α-tubulin isoforms and their post-translational modifications will be required.

### 5.3. Cytoskeleton Rearrangement in the Regulation of Cancer Metabolism

CSCs exhibit elevated mitochondrial fusion with a metabolic rewiring to OXPHOS [145,146] and a rearranged cytoskeleton network. Cytoskeletal rearrangement is essential for promoting cell movement, invasion, and proliferation. Emerging evidence is beginning to highlight that cytoskeleton rearrangement somehow changes cellular bioenergetics. In yeast, decreased actin turnover, leading to accumulation of aggregates of F-actin, triggers an increase in the production of ROS by mitochondria [153,154]. Some evidence implies a role for actin in ROS production and mitochondrial clustering following cell death signaling [155]. Recently, Hu et al. revealed a mechanism by which PI3K signaling promotes glycolysis. Aldolase functions not only as a key metabolic enzyme for glycolysis but also interacts with cytoskeletal components that control actin polymerization [156]. Activation of the PI3K pathway allows for the physical dissociation of aldolase from F-actin into the cytoplasm, where it is active. These data show an AKT-independent role of PI3K in altering glycolysis through spatial redistribution of aldolase via cytoskeletal rearrangement. These data suggest a mechanism of activating a metabolic enzyme through rearrangement of the cytoskeleton to increase the metabolic flux. Amelia et al. demonstrated that βIII-tubulin promotes the reduction of the reliance of lung cancer cells on glycolysis when glucose becomes limiting. When glucose supply is abundant, βIII-tubulin reduces the reliance of tumor cells on glucose metabolism and promotes responsive AKT signaling to prime cells to adapt when glucose becomes lacking. BIII-tubulin protects tumor cells from ER stress in nutrient-starved conditions via modulation of glycolytic dependency [120]. Recently, the cytoskeletal protein syntaphilin (SNPH), known for arresting mitochondrial movement at sites of high energy demands [157], was found to be expressed in cancer [10,157]. In addition, a recent study revealed that a novel isoform of SNPH maintains the activity of ETC II and local tumor growth. Thus, SNPH downregulation increases mitochondrial superoxide production and reduces cell proliferation [158]. Therefore, the regulatory mechanism between cytoskeletal rearrangement or cytoskeletal regulators and cellular bioenergetics is crucial for understanding tumor cells in response to diverse stimuli.

### 5.4. Cytoskeleton-Mitochondria Interplay Regulates EMT

A growing body of evidence points to the functional interaction between the cytoskeleton and mitochondria as a crucial regulatory center. Interestingly, mitochondria, which are affected by autophagy-related processes, define the energy supply that cancer cells use to reorganize the cytoskeleton and sustain cell movement during EMT [5,159,160,161]. EMT is firstly determined by morphological reprogramming of the cellular architecture, which is guided by changes in the interaction properties of cells with the surrounding microenvironment and is supported by a reorganized cytoskeleton [162,163]. Accumulation of mitochondria near the cell membrane is critical to promote the formation of filopodia, lamellipodia, and other cellular structures for motility during EMT [160,161]. The involvement of ROS in regulating EMT via actin reorganization has been suggested in recent years. Upon exposure to temporary ROS, oxidized actin filaments could promote the contractility of actomyosin by inducing the disassembly of actin and myosin, leading to the formation of stress fibers and the promotion of cell spreading [164]. At the leading edge, mitochondria support enhanced cell motility and invasion through providing a local source of energy. The importance of regional ATP production for mitochondrial repositioning at the leading edge was demonstrated by using tumor cells that contain OXPHOS-deficient mitochondria (called ρ^0^ cells). ρ^0^ cells failed to reposition mitochondria to focal adhesion complexes [5,165]. Furthermore, Max-Hinderk Schuler et al. demonstrated that Miro-1-mediated mitochondrial repositioning at the leading edge provides localized energy demands to promote cell migration by supporting membrane protrusion and focal adhesion stability [165]. Another independent study that relied on misfolding of the ETC II subunit SDHB by treatment with the mitochondrial HSP90 inhibitor Gamitrinib prevented accumulation of mitochondria to focal adhesion. In addition, pharmacological inhibition of ETC I, III, and V (rotenone, antimycin A, and oligomycin) inhibited mitochondrial repositioning to the cortical cytoskeleton [166]. The same group showed that under nutrient deprivation conditions, cancer cells protect cytoskeletal dynamics and motility through the chaperone function of mitochondria-associated HSP90. Increased expression of HSP90 preserves residual OXPHOS and ATP production, preventing mitochondrial disruption [167]. Mutations in mtDNA content are also associated with EMT in cancer cells. Studies using NSCLC cells showed that EMT induced by TGF-β leads to an increase in mtDNA copy number [168]. Increased mtDNA is associated with a higher energy requirement for EMT and with metastasis. Indeed, mitochondrial localization through cytoskeleton provides a means to concentrated energy where is most needed.

## 6. Therapeutic Strategies Targeting CSCs

Improvement of anticancer therapeutic strategies is often limited by incomplete knowledge of the molecular mechanisms underlying cancer biology and cellular metabolism in response to treatment. CSCs exhibit elevated mitochondrial fusion with a metabolic rewiring to OXPHOS [169,170] and a rearranged cytoskeleton network. Increased mitochondrial fusion promotes ATP production via OXPHOS, allowing the cancer cells to survive without energy constraints. A list of inhibitors under study in vivo and in the clinic is shown in Table 1 and Figure 1.

### 6.1. Targeting Mitochondrial Dynamics

Inhibition of mitochondrial fusion is a promising strategy because it plays extensive roles in cancer biology. Changes in levels of mitochondrial fusion proteins MFN 1 and 2 affect mitochondrial morphology and integrity. MFN 1- and 2-deficient cells are characterized by elevated mitochondrial fragmentation with a loss of mitochondrial membrane potential and defects in mitochondrial respiration [79]. Moreover, MFN2 depletion downregulates genes that participate in OXPHOS complex, such as ETC I (p39), II (p70), III (p49,) and V (α subunit), leading to a decrease in their enzymatic activity [94,172]. A selective inhibitor of MFN1, Beta II protein kinase C (βIIPKC) accumulates on the mitochondrial outer membrane and phosphorylates MFN1 at serine 86. Phosphorylation of MFN1 induces a partial loss of its GTPases activity and results in buildup of fragmented and dysfunctional mitochondria [173]. Thus, examining other MFN- and OPA1-targeting molecules could further improve our understanding of the role of mitochondrial dynamics in CSC maintenance from a cellular bioenergetics perspective.

### 6.2. Targeting OXPHOS

OXPHOS inhibitors are promising in targeting tumors with increased dependency on OXPHOS. One of the most promising OXPHOS inhibitors is metformin, a Food and Drug Administration (FDA)-approved diabetic drug repurposed for cancer treatment. By inhibiting ETC I, metformin reduces OXPHOS and tumor growth in multiple cancer models [174,175]. Phenformin, an alternative to metformin, is another biguanide that is readily transported into cancer cells and shows a higher affinity to the mitochondrial membrane than metformin [175]. In non-small cell lung cancer (NSCLC), ALDH-positive cells exhibit CSC phenotypes such as self-renewal and multidrug resistance. An ALDH inhibitor, gossypol reduces ATP production with a decrease in NADH levels in NSCLC. A new metformin derivative, IM156, targets slow-cycling tumor cells that tend to escape from treatment with conventional chemotherapies that have been designed to target fast-cycling cells. In addition, IM156-mediated inhibition of mitochondrial function prevents the growth of oral cancer and glioblastoma [176,177]. Although the above-mentioned therapeutics have been demonstrated to reduce mitochondrial function and survival of CSCs, it should be considered that inhibition of a single specific component of OXPHOS and ATP production may induce alternative pathways, precluding the efficacy of regimens to ultimately eradicate CSCs.

### 6.3. Targeting the Cytoskeleton

The coupling between mitochondria and the cytoskeleton in the regulation of mitochondrial dynamics and respiration can be a novel target for CSCs. Nowadays, microtubule-targeted agents (MTAs) constitute a class of anticancer drugs largely used in the clinic. Among them, taxanes and vinca alkaloids are powerful inhibitors of microtubule dynamics and are used to treat multiple malignancies. Several studies show that cytochalasin D stimulates actin depolymerization and downregulates mitochondrial function via VDAC closure. Koyama et al. demonstrated that inhibiting Rho-kinase with Y-27632, diminishes ATP production [178]. Paclitaxel and colchicine, which target β-tubulin, increase the mitochondrial membrane potential by decreasing free cytosolic tubulin, which leads to opening of VDAC. This process may explain chemoresistance against tubulin-targeted agents. Because cytoskeletal rearrangement contributes mostly to maintaining quality control and stress resistance of cancers, investigating possible combinations with mitochondria-targeted drugs or synthetic lethal targets may be a promising strategy for overcoming drug resistance in CSCs.

## 7. Conclusions

In this review, we summarized the mechanisms regulating mitochondrial dynamics and functions that are modulated by the interaction between mitochondria and the cytoskeleton (Figure 3). All these interactions play important roles in the regulation of mitochondrial dynamics, tempo-regional repositioning, re-arrangement, and energy fluxes in the cell, demonstrating system-level properties of coordinating cellular bioenergetics and cellular behavior. In particular, fatty acid oxidation, one of the major characteristics of CSCs, is also regulated by mitochondrial dynamics in response to cytoskeletal remodeling. This metabolic preference is a promising target for certain types of cancer with elevated fatty acid utilization. Many recent studies have demonstrated that mitochondria-centric metabolism, OXPHOS, is upregulated in a variety of cancers, potentially rendering them sensitive to mitochondria-targeted drugs. Cancer’s intrinsic sensitivity to mitochondria-targeted drugs should continue to be characterized, environmental drivers of cancer cell susceptibility to these drugs must be recognized, and combinations with cytoskeleton-targeted drugs should be evaluated. Tumor heterogeneity may also affect cytoskeletal protein expression and organization to utilize cellular dynamics and coordination. Thus, understanding the mechanisms of mitochondrial fitness, including the impact of the cytoskeletal contribution to extrinsic and intrinsic stress responses, can facilitate the development of novel therapeutic strategies that directly eradicate CSCs in patients with drug-resistant tumors.

## Figures and Tables

**Figure 1 cells-09-01691-f001:**
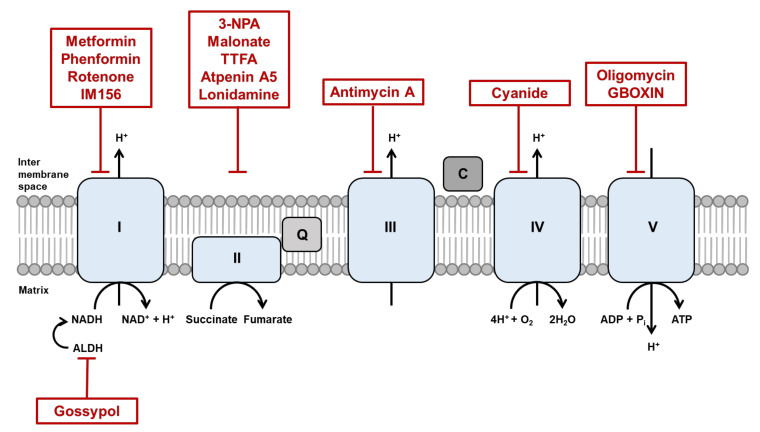
Inhibitors of oxidative phosphorylation (OXPHOS). The OXPHOS metabolic pathway generates ATP by transporting electrons through a series of transmembrane complexes in the mitochondrial inner membrane, known as the electron transport chain (ETC). Electrons flow through complex I, complex II, coenzyme Q (Q), complex III, cytochrome c (C), and complex IV, with O_2_ acting as the terminal electron acceptor. Therapeutic agents that are OXPHOS inhibitors are shown in red boxes. 3-NPA: 3-nitropropionic acid; TTFA: 2-thenoyltrifluoroacetone.

**Figure 2 cells-09-01691-f002:**
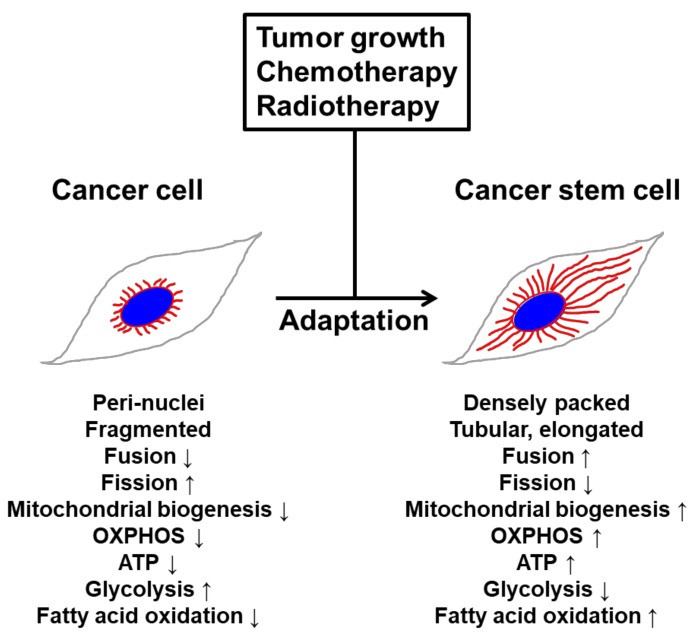
A simplified scheme of mitochondrial dynamics in cancer cells and cancer stem cells (CSCs). In cancer cells, mitochondria are usually localized in the nuclear periphery and characterized by sphere-like, fragmented, and punctate morphology. Correspondingly, mitochondrial fission is high, whereas mitochondrial biogenesis is low. Cancer cells generally rely on glycolysis as the major energy source and have low levels of OXPHOS, ATP production, and fatty acid oxidation levels. In CSCs, mitochondria exhibit more elongated and enlarged tubular morphology. Correspondingly, mitochondrial fusion and biogenesis increase with the densely packed mitochondria. Comparably, CSCs have higher OXPHOS, ATP production, and fatty acid oxidation levels. Blue: nucleus; red lines: mitochondria.

**Figure 3 cells-09-01691-f003:**
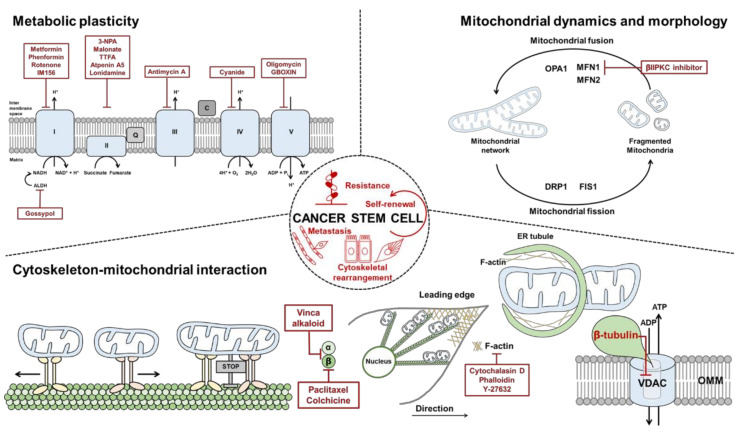
Schematic overview of therapeutic targeting of cancer stem cells (CSCs). Therapeutic approaches (indicated in red font) against CSCs may include reduction of metabolic plasticity, pharmacological targeting of mitochondrial dynamics and morphology, and cytoskeleton-mitochondria interaction-targeted drugs. For detailed mechanisms see text.

**Table 1 cells-09-01691-t001:** List of Inhibitors under study in vivo or in the clinic as anticancer therapeutics.

Categories	Compound	Target	Conditions	Clinical Trials	Clinical Trials Identifier/Ref.
Mitochondrial dynamics and function	βIIPKC inhibitor	MFN1	Heart failure	Experimental	[156,157]
	ONC201	Mitochondrial respiration	Multiple cancers	Several trials in process	[171]
NADH production	Gossypol	Aldehyde dehydrogenase	Multiple cancers	Several trials in process	[158]
Microtubule	Paclitaxel	β-Tubulin subunit	Multiple cancers	Several trials in process	[159,160]
	Colchicine	β-Tubulin subunit	Multiple cancers	Several trials in process	[161,162]
	Vinca alkaloid	Tubulin dimer	Multiple cancers	Several trials in process	[163,164]
Actin	Cytochalasin D	Actin monomer	Infertility	Preclinical	NCT03677492
	Phalloidin	F-actin	Depression	Preclinical recruiting	NCT04137458
	Y-27632	Rho-associated kinase	Hippocampal neurons	Preclinical	[165]
ETC	Metformin	Complex I	Multiple cancers	Several trials in progress	NCT03477162
	Phenformin	Complex I	Diabetes mellitus, type 2	Preclinical completed	NCT02475499
	Rotenone	Complex I	Parkinson’s disease	Preclinical enrolling	NCT04287543
	IM156	Complex I	Advanced solid tumor and lymphoma	Phase I active	NCT03272256
	3-NPA	Complex II	Parkinson’s disease	Preclinical	[166]
	Malonate	Complex II	Parkinson’s disease	Phase I terminated	NCT01476085
	TTFA	Complex II	Cancer	Experimental	[167]
	Atpenin A5	Complex II	Cardiac ischemia-reperfusion injury	Preclinical	[168]
	Lonidamine	Complex II	Symptomatic benign prostatic hyperplasia, enlarged prostate	Phase 2, 3 terminated	NCT00237536, NCT00435448
	Antimycin A	Complex III	Lung cancer	Experimental	[169]
	Cyanide	Complex IV	Malignant glioma	Phase 2 recruiting	NCT00075387
	Oligomycin	Complex V	Cancer cachexia, transthyretin amyloidosis	Preclinical recruiting	NCT03144128, NCT03328338
	GBOXIN	Complex V	Glioblastoma	Preclinical	[170]

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
