# Peer review of "Role of Mitochondria-Cytoskeleton Interactions in the Regulation of Mitochondrial Structure and Function in Cancer Stem Cells"

_cells, 2020, doi:10.3390/cells9071691_

Round 1
Reviewer 1 Report
The topic about regulation of mitochondrial structure and function in cancer stem cells and the connections between mitochondrial organization and cancer metabolic plasticity are very important. The mechanisms of mitochondrial fitness, including the impact of cytoskeletal contribution to extrinsic and intrinsic stress responses, can facilitate the development of novel therapeutic strategies. The article is interesting and easy to read. However, some questions arose:
- What did you had in mind “the increased mitochondrial mass “. In which types of tumors has it been found?
- Is there anything more specific known about the relationship between CSC and the cytoskeleton?
- What else could affect the of VDAC permeability besides tubulin?
- What is known about the toxicity of the inhibitors of mitochondrial metabolism under study in vivo?
- Could the cancer stem cell markers (CD44, CD133, EpCAM, ALDH) also be present in normal tissues?
- Could the adenylate kinase also participate in regulation of the mitochondrial function by actin? What are the relationships between non-actin cytoskeleton and VDAC permeability in CSC?
Author Response
Thank you for considering our article for publication in Cells. We are grateful to the reviewers for the valuable suggestions provided. Below are the responses to each of the reviewers’ comments.
- What did you had in mind “the increased mitochondrial mass “. In which types of tumors has it been found?
Response: We have used the phrase “the increased mitochondrial mass” to indicate structurally and functionally competent mitochondria that can be assessed by canonical methods. Most published data on mitochondrial mass rely on flow cytometric analysis after staining with MitoTracker. Mitotracker passively diffuses across the plasma membrane and accumulates in the mitochondrial inner membrane, suggesting that the fluorescence observed after staining is correlated with the full mitochondrial mass and inner membrane quantity. The concentration of this dye is ∼300-fold higher in the mitochondrial matrix than in the medium surrounding these organelles, which may be because of the pH of the inner mitochondrial membrane. Mitochondrial mass can also be analyzed by transmission electron microscopy (TEM). Fragmented mitochondria exhibit accumulated electrons that appear as dark spots with swollen and unclear cristae, whereas elongated mitochondria exhibited less electron accumulation and appear as less dark with finely divided cristae. For example, pancreatic cancer cells that survived KRAS-ablation exhibited an increased mitochondrial mass in TEM analysis (Nature. 2014 Oct 30;514(7524):628-32. PMID: 25119024). Björn von Eyss et al. demonstrated MYC-driven elevation of the mitochondrial mass in breast cancer (Cancer Cell. 2015 Dec 14;28(6):743-757, PMID: 26678338). Furthermore, MYC and MCL induce elevations in the mitochondrial mass in breast cancer as assessed by TEM analysis (Cell Metab. 2017 Oct 3;26(4):633-647, PMID: 28978427).
- Is there anything more specific known about the relationship between CSC and the cytoskeleton?
Response: Several researchers have demonstrated that CSCs can be characterized into the following three categories: anchorage-independent survival (Science. 2001 Sep 7;293(5536):1829-32, PMID: 11546872, Biochim Biophys Acta. 2009 Dec;1796(2):75-90, PMID: 19306912), resistant to chemotherapy (Anticancer Res. Sep-Oct 1997;17(5A):3393-401, PMID: 9413178, Sci Rep. 2018 Aug 9;8(1):11935, PMID: 30093656, Eur Rev Med Pharmacol Sci. Jan-Feb 2009;13(1):13-21, PMID: 19364082), and metastasis- and relapse-prone phenotypes (Cell Research 1, 141–151(1990), Nat Rev Mol Cell Biol. 2003 Aug;4(8):657-65, PMID: 12923528, Br J Pharmacol. 2014 Dec;171(24):5507-23, PMID: 24665826, Cancer Res. 2015 Jan 1;75(1):203-15, PMID: 25503560, J Cancer Res Clin Oncol. 2018 Nov;144(11):2195-2205, PMID: 30094535). These phenotypes are closely associated with and governed by the cytoskeletal architecture. In glioblastoma, the actin cytoskeleton regulator Arp2/3 complex stimulates glioma initiating cell (glioblastoma stem cell) motility and cell migration, thereby triggering tumor invasion. (Oncotarget. 2017 May 16; 8(20): 33353–33364, PMID: 28380416). Metastatic and basal-like breast cancer cells exhibited acetylated alpha tubulin and high migration and invasion (Cancer Res. 2015 Jan 1;75(1):203-15. PMID: 25503560). A thorough search for more specific references using the key words cancer stem cells and cytoskeleton with customized queries was conducted; these included cancer stem cells and actin (hits 233), cancer stem cells and microtubule (hits 157), cancer stem cells and intermediate filament (hits 43) on July 3rd in 2020. Most studies demonstrated a relationship between the cytoskeleton and EMT-metastasis in CSC model systems because of the role of the cytoskeleton in the maintenance and regulation of the cellular architecture (For example, Sci Rep. 2017 Apr 13;7:46312, PMID: 28406185, J Oncol, 2011;2011:591427, PMID: 21253528). More specifically, we have added the key words mitochondria with the same queries: cancer stem cells and actin (hits 4), cancer stem cells and microtubule (hits 4), cancer stem cells and intermediate filament (hits 2) on the same day. Shagieva et al. demonstrated that depletion of ROS downregulates EMT in cervical cancer cells (Oncotarget. 2017 Jan 17;8(3):4901-4913, PMID: 27902484). This report also correlated the EMT phenotype and not specifically mitochondrial function or dynamics, which we have discussed in this manuscript. There is a gap in information regarding the relationship between the cytoskeleton and CSC, especially where mitochondrial functional augmentation is concerned; this information is important as it can give the CSCs a survival advantage.
- What else could affect the of VDAC permeability besides tubulin?
Response: The permeability of VDAC can be affected by the hexokinase-VDAC interaction. In tumor cells with elevated levels of hexokinase bound to VDAC, apoptosis was suppressed and proliferation was facilitated (J Biol Chem. 2008 May 9;283(19):13482-90, PMID:18308720). Furthermore, detachment of hexokinase from VDAC can be modulated by GSK3-ęžµ-mediated phosphorylation, and phosphorylation of VDAC by PKC-ε (PKC-ε) promotes hexokinase binding (Circ Res. 2003 May 2;92(8):873-80, PMID: 12663490, Am J Physiol Heart Circ Physiol. 2007 Oct;293(4):H2056-63, PMID: 17675573). In addition, overload of Ca2+ ions into the mitochondria would lead to the accumulation of mitochondrial superoxide anions, ultimately facilitating Ca2+-induced irreversible opening of the VDAC, leading to apoptosis and necrotic cell death (Cell Calcium. 2011 Sep;50(3):222-33. PMID: 21601280).
- What is known about the toxicity of the inhibitors of mitochondrial metabolism under study in vivo?
Response: VLX600, a recently designed iron chelator, inhibits mitochondrial respiration, bioenergetic failure and cell death. Inhibition of mitochondrial respiration using VLX600 reduced the growth of a colon tumor xenograft growth in mice (Nat Commun. 2014;5:3295, PMID: 24548894). In this report, significant antitumor activity was demonstrated in the dose range 0.5–16 mg/kg. Minimal toxicity was observed at doses up to 4.5 mg/kg. At higher doses, local intolerance problems at the injection site were observed, and mice treated with the highest doses suffered from tremor or decreased motor activity and nervous behavior a few minutes post-bolus injection. A phase I trial of VLX 600 in patients with refractory advanced solid tumors is underway (NCT02222363, Invest New Drugs. 2019 Aug;37(4):684-692, PMID: 30460505). Another new small-molecule inhibitor of mitochondrial metabolism is Gboxin, which interacts with OXPHOS complexes and thereby inhibits complex V. It decreases the growth of glioblastoma cells in vitro and in vivo. The authors demonstrated that Gboxin exerts its tumor cell specific toxicity in normal cells in primary culture and in vivo (Nature. 2019 Mar; 567(7748): 341–346, PMID: 30842654).
- Could the cancer stem cell markers (CD44, CD133, EpCAM, ALDH) also be present in normal tissues?
Response: We have added references describing the expression of CD44 in keratinocytes (Histochem Cell Biol (2012) 137:107–120, PMID: 22072421), CD133 in normal skin cells (Biomed Res Int. 2013;2013:385604, PMID: 24102054), EpCAM in normal liver cells (Cell Stress. 2019 May 21;3(6):165-180, PMID: 31225512), and ALDH in normal mammary tissues in the manuscript (Cell Stem Cell. 2007 Nov;1(5):555-67, PMID: 18371393).
- Could the adenylate kinase also participate in regulation of the mitochondrial function by actin? What are the relationships between non-actin cytoskeleton and VDAC permeability in CSC?
Response: Based on literature review, the participation of actin cytoskeleton in regulating mitochondrial function is not well-known. On searching references using the key words adenylate kinase and actin, 109 results were found but cross-searching with mitochondrial function revealed no hits on July 3, 2020. The relationships between non-actin cytoskeleton such as intermediate filaments and VDAC permeability are not well-understood. However, regulation of VDAC permeability via class II ęžµ-tubulin and hexokinase (HK) complex occurs in malignant tumor cells. Yuan et al. demonstrated that a VDAC serine phosphorylation residue, Ser-103 on the VDAC cytosolic side interface in both VDAC1 folding patterns, may be a target for hexokinase (FASEB J. 2008 Aug;22(8):2809-20, PMID: 18381814). (Cells. 2019 Mar; 8(3): 239, PMID: 30871176, Oncotarget. 2019 Feb 26; 10(17): 1606–1624, PMID: 30899431).
Reviewer 2 Report
In their review, Jungmin Kim and Jae-Ho Cheong highlight the central role of mitochondrial function in cancer stem cells. Specifically, they underline how mitochondria and the cytoskeleton interact, what are the functional consequences and propose to target this interaction as a potential therapeutic approach. Overall, this review is well written and adds a new perspective to the field of cancer stem cells (CSCs) and mitochondria. However, certain concerns need to be addressed:
- In the past 2-3 years there were multiple reviews published about cancer stem cells and mitochondria. The introduction section will benefit from including a clear statement regarding the distinction of the current review compared to the others and proper reference to other reviews on the topic.
- The authors frequently use the term “normal/regular cancer cells” both in the text and the figure 2 to distinguish them from CSCs. This is unnecessary and at times, confusing.
- Lines 178-182 lack proper reference and the message is not clear. Both fission and fusion are necessary for the maintenance of healthy mitochondria populations. KO models of either fission or fusion proteins resulted in mitochondrial abnormal morphologies and functions.
- In section 3.3 the authors discuss mitochondrial structure-function in CSCs. Although they bring up opposing effects of different oncogenes (i.e Erk result in fragmented mitochondria and Myc in fused) they do not discuss the reasons. This section will benefit from an in-depth discussion of the opposing consequences, is it due to the microenvironment? How does it fit with studies done in CSCs?
- Some drugs that affect mitochondria structure and function are currently being tested in early phase clinical trials and should be included in section 3.3.
- Greer et al. 2018 ONC201 kills breast cancer cells in vitro by targeting mitochondria.
- Graves et al. 2019 Mitochondrial Protease ClpP Is a Target for the Anticancer Compounds ONC201 and Related Analogues.
- The opening paragraph of section 4 lacks references.
- In section 5.4, you need to add references related to EMT.
- This paper should translocation of mitochondria to the leading edge of migrating cells and should be included: Miro1-mediated Mitochondrial Positioning Shapes Intracellular Energy Gradients Required for Cell Migration. Max-Hinderk Schuler et. al. Mol.Biol.Cell 2017
- The final paragraph in section 5.4 discussing the extracellular matrix and YAP seems a little off topic. The link to the overall frame of this review should be better explained.
- Lastly, the cytoskeleton is targeted to treat cancer for many years (Taxol for example), it is therefore important to address the question how is this going to be any different?
Minor:
Line 88, extra space
Line 96, what are JARID1B high cells?
Line 116, “extremely of help” should be extremely helpful or useful.
Line 125, “intimately connected” not clear, maybe transferred?
Line 177, not clear what you mean by “cell mass”
Lines 309 and 332 the term “ANT” is not defined
Line 335 probably should be interaction instead of “interacts”
Line 354, “abundant” doesn’t make sense
Line 389, is the word “content” needed here?
Author Response
Thank you for considering our article for publication in Cells. We appreciate the reviewers’ comments and suggestions, which helped to greatly improve our manuscript. Our point-by-point response are provided in a separate file. Altered text in the manuscript has been highlighted in yellow.
1.In the past 2-3 years there were multiple reviews published about cancer stem cells and mitochondria. The introduction section will benefit from including a clear statement regarding the distinction of the current review compared to the others and proper reference to other reviews on the topic.
Response: We have revisited previous reviews on cancer stem cells and mitochondria. Using the key words with cancer stem cell and mitochondria, 676 hits were found in the past 5 years. More specifically, by cross-searching with the key word cytoskeleton, only 3 hits were found (Oncotarget. 2017 Jan 17;8(3):4901-4913, PMID: 27902484, Mol Oncol. 2016 Aug;10(7):949-65, PMID: 27106131, Hum Gene Ther. 2019 Mar;30(3):365-377, PMID: 30266073). We have revised the Introduction section and the related main narrative in the manuscript.
2.The authors frequently use the term “normal/regular cancer cells” both in the text and the figure 2 to distinguish them from CSCs. This is unnecessary and at times, confusing.
Response: We have deleted terms “normal and regular” from the manuscript, Figure 2, and figure legend.
3.Lines 178-182 lack proper reference and the message is not clear. Both fission and fusion are necessary for the maintenance of healthy mitochondria populations. KO models of either fission or fusion proteins resulted in mitochondrial abnormal morphologies and functions.
Response: We have added the proper references to the manuscript. Regarding mitochondrial fission, we have added the relevant content and revised related passages. Briefly, mitochondrial fission is crucially involved in the maintenance of healthy mitochondrial populations and mitochondrial inheritance by daughter cells during cellular division. Mitochondrial fragmentation occurs in response to various factors such as intracellular calcium levels and ATP availability (Mol Cell Biol. 2003 Aug;23(15):5409-20, PMID: 12861026, Mol Cell Biochem. 2005; 272(1–2):187–199, PMID: 16010987). For example, the calcium-sensitive phosphatase calcineurin promotes fission by dephosphorylating cytosolic Drp1, which causes translocation to the mitochondria (Cell Death Differ. 2010; 17(11):1785–1794, PMID: 20489733). Plasticity bestows the adaptive flexibility required to adjust and adapt to changing cellular stresses and metabolic demands. With respect to plasticity of mitochondrial dynamics, we suggest that constant network-like remodeling establishes a mechanism for the quality control of the mitochondrial population with important ramifications for gain of fitness of CSCs. It is well-documented that mitochondrial fusion during nutrient starvation facilitates fatty acid trafficking and oxidation for survival. Furthermore, mitochondrial fusion causes an increase in the mitochondrial cristae number, which is associated with dimerization of the ATP synthase and thereby increases ATP synthesis activity (Nat Cell Biol 2011, 13, 589-598, PMID: 21478857). Therefore, we focused on mitochondrial fusion in the fitness of CSCs against diverse stresses that may be encountered.
4.In section 3.3 the authors discuss mitochondrial structure-function in CSCs. Although they bring up opposing effects of different oncogenes (i.e Erk result in fragmented mitochondria and Myc in fused) they do not discuss the reasons. This section will benefit from an in-depth discussion of the opposing consequences, is it due to the microenvironment? How does it fit with studies done in CSCs?
Response: We have revised the text stating that the opposing effects may be modulated by different capacities for nutrient utilization from a poor tumor microenvironment. The ERK pathway induces glycolysis, whereas MYC signaling utilizes fatty acid oxidation. We have proposed that cancer cells rely on glycolysis with fragmented mitochondria, whereas CSCs have higher mitochondria-centric OXPHOS with network-like mitochondria. These different phenotypes can be regulated by sophisticated cooperation between mitochondrial dynamics and energy substrate availability in the tumor microenvironment. We have also added the correlation between the regulation of ERK and MYC in the PDAC model system (Cancer Cell. 2018 Nov 12; 34(5): 807–822.e7., PMID: 30423298).
5.Some drugs that affect mitochondria structure and function are currently being tested in early phase clinical trials and should be included in section 3.3.
Greer et al. 2018 ONC201 kills breast cancer cells in vitro by targeting mitochondria.
Graves et al. 2019 Mitochondrial Protease ClpP Is a Target for the Anticancer Compounds ONC201 and Related Analogues.
Response: We have added content on ONC201 and its reference in the manuscript.
6.The opening paragraph of section 4 lacks references.
Response: We have added the proper references to the revised manuscript.
7.In section 5.4, you need to add references related to EMT.
Response: We have added the proper references to the revised manuscript.
8.This paper should translocation of mitochondria to the leading edge of migrating cells and should be included: Miro1-mediated Mitochondrial Positioning Shapes Intracellular Energy Gradients Required for Cell Migration. Max-Hinderk Schuler et. al. Mol.Biol.Cell 2017: Added in manuscript
Response: We have added a description of Miro-1 and its reference to the revised manuscript.
9.The final paragraph in section 5.4 discussing the extracellular matrix and YAP seems a little off topic. The link to the overall frame of this review should be better explained.
Response: Based on your suggestion, we have removed the paragraph describing ECM and YAP from the revised manuscript.
10.Lastly, the cytoskeleton is targeted to treat cancer for many years (Taxol for example), it is therefore important to address the question how is this going to be any different?
Response: Cytoskeleton targeted drugs are widely used in cancer treatment, but limitations remain such as resistance and relapse. Vinca alkaloids bind to binding sites on tubulin that are distinct from those of taxanes and colchicine. Vinca alkaloids bind the α/ęžµ-tubulin heterodimer and inhibit microtubule elongation. In contrast, taxol binds to ęžµ-tubulin and stabilizes the microtubule. Tubulin was found to be strongly associated with mitochondrial membranes in both purified organelles and whole cells (J Biol Chem. 2002 Sep 13;277(37):33664-9, PMID: 12087096, J Biol Chem. 2005 Jan 7;280(1):715-21, PMID: 15516333). The mitochondrial tubulin sub-fraction is enriched in class III ęžµ-tubulin (TUBB3), which may explain the insensitivity to taxol (Mol Cancer Ther. 2008 Jul;7(7):2070-9, PMID: 18645017). In addition, class III ęžµ-tubulin (TUBB3) overexpression is a marker of resistance to taxol in vitro, in vivo, and in the clinic (Lung Cancer. 2010 Feb;67(2):136-43, PMID: 19828208, Int J Cancer. 2007 May 15;120(10):2078-85, PMID: 17285590). Tubulin is encoded by a multigene family that produces at least 8 distinct isotypes for each of the α- and ęžµ- tubulin proteins (Int Rev Cytol. 1998;178:207-75, PMID: 9348671, Cytoskeleton (Hoboken). 2010 Apr;67(4):214-23, PMID: 20191564). Therefore, an improved understanding of drug action and resistance has important implications for chemotherapy using microtubule-targeting agents. Notably, paclitaxel (PTX) also binds to HSP90 in the C-terminal domain; in contrast to traditional HSP90 inhibitors, PTX has a stimulating effect on HSP90 (Curr Med Chem. 2008;15(26):2702-17, PMID: 18991631). In agreement with the previous report, Chavez et al. recently demonstrated that PTX alters mitochondrial respiration and ATP synthase structure. Eventually, PTX-driven conformational changes improve mitochondrial function, particularly the maximum respiratory capacity (Chavez et al., 2019, Cell Reports 29, 2371–2383, PMID: 31747606). Taken together, taxol binds to ęžµ-tubulin and stabilizes the microtubule while interacting with and stimulating HSP90. In addition, PTX can modulate mitochondrial function by causing conformational changes in ATP synthase. This may be important information for fully understanding the insensitive response to PTX and mitochondrial functional augmentation after the chemotherapy.
Minor:
Line 88, extra space
Response: We have corrected line 88.
Line 96, what are JARID1B high cells?
Response: Roesch et al. identified a slow-cycling cell subpopulation in melanoma based on the expression of histone 3 K4 demethylase (JARID1B). These subpopulations exhibited a treatment-resistant phenotype and increased mitochondrial respiratory capacity (Cancer Cell. 2013 Jun 10;23(6):811-25. PMID: 23764003). We have added this study to the reference list in the revised manuscript.
Line 116, “extremely of help” should be extremely helpful or useful.
Response: We have corrected this error.
Line 125, “intimately connected” not clear, maybe transferred?
Response: We have changed “intimately connected” to “involved in”.
Line 177, not clear what you mean by “cell mass”
Response: Miettinen et al. demonstrated that increased mitochondrial connectivity may alleviate energy transport limitations, enabling a higher metabolic rate and larger cell size (Trends Cell Biol. 2017 Jun;27(6):393-402. PMID: 28284466).
Lines 309 and 332 the term “ANT” is not defined
Response: We have added the definition of ANT, adenine nucleotide translocator, to the manuscript.
Line 335 probably should be interaction instead of “interacts”
Response: We have revised this point.
Line 354, “abundant” doesn’t make sense
Response: We have reviewed the references again and changed this term to “limiting”.
Line 389, is the word “content” needed here?
Response: We have deleted this term.